# Determining the drivers for snow gliding

Reinhard Fromm[1], Sonja Baumgärtner[2], Georg Leitinger[2], Erich Tasser[3], Peter Höller[1]

[1]Federal Research and Training Centre for Forests, Natural Hazards and Landscape – BFW, Department of Natural Hazards, Innsbruck, 6020, Austria

5  [2]Department of Ecology, University of Innsbruck, Innsbruck, 6020, Austria

[3]Institute for Alpine Environment, Eurac research, Bozen, 39100, Italy

*Correspondence to*: Reinhard Fromm (reinhard.fromm@bfw.gv.at)

10  **Abstract.** Snow gliding is a key factor for snow glide avalanche formation and soil erosion. This study considers atmospheric and snow variables, vegetation characteristics, and soil properties, and determines their relevance for snow gliding at a test site (Wildkogel, Upper Pinzgau, Austria) during winter 2014/15. The time-dependent data were collected at a high temporal resolution. In addition to conventional sensors a 'snow melt analyzer' was used.

The analysis shows that the soil temperature 10 cm below the surface, the phytomass of mosses, the liquid water content in 15  the snowpack, and the static friction coefficient of the glide shoes had significant influence on snow gliding during the whole winter. In the first period (October to January) the soil moisture at the surface and 1.5cm below the surface and the length of the slope uphill the glide shoes affected the snow gliding, too. In the second period (February to May) the soil temperature at the surface, the soil moisture 10cm below the surface, and the slope angle had additional influence on snow gliding.

The role of the vegetation in the snow glide process is determined by the influence on the static friction coefficient caused by 20  its composition and characteristics and that moss-rich and short-stemmed canopies seem to be more interconnected with the snowpack.

Additional to the soil and snow properties, the topography and the vegetation characteristics, further investigations may be focused on the freezing and melting processes in the uppermost soil layers, and at the soil surface.

## 1 Introduction

25  Deposited snow on the ground is in motion caused by gravity, external forces, or metamorphism. The movement inside the snowpack is called creeping, and the sliding of the entire snowpack on an inclined ground surface is referred as snow gliding (In der Gand and Zupancic, 1966). Snow gliding is favored by a smooth ground surface and a lowermost layer of wet snow (In der Gand and Zupancic, 1966). Once the glide motion turns into an avalanche movement, the process is called a glide avalanche (UNESCO, 1981).

30  The presence of liquid water at the bottom of the snowpack is a basic requirement for snow gliding (In der Gand, 1954; Lackinger, 1988; McClung et al., 1994; Mitterer and Schweizer, 2013). Several sources exist to provide liquid water to this location (Ceaglio et al., 2012; Ceaglio et al., 2017; Mitterer and Schweizer, 2012). Rain on the snow surface, as well as melting snow near to the surface (Koh and Jordan, 1995), can percolate the isothermal snowpack. Geothermal heat flux can provide energy to melt snow at the bottom of the snowpack (McClung and Clarke, 1987). The suction head can lift water (Mitterer and 35  Schweizer, 2012; Ceaglio et al., 2017) which is produced by melting ice stored in the soil or it can be advected through channels in the soil (ground water outflow).

In addition to the presence of liquid water at the bottom of the snowpack, further variables influence the intensity of snow gliding. Therefore, air temperature can be used to classify the glide snow avalanches into warm-temperature events and cold-temperature events (Clarke and McClung, 1999). The viscosity of snow depends on the snow temperature (Loth et al., 1993; 40  Morris, 1994) and snow water content (Mitterer and Schweizer, 2012; McClung and Clarke, 1987). The slope angle, the micro relief, and the hydrological properties of the slope influence the glide velocity (Ceaglio et al., 2017; McClung and Schaerer,

1999; Margreth, 2007). Friction originated by the vegetation depends on its composition and height (Höller et al., 2009). Both the vegetation and the micro relief depend on the land use, which is an input for snow glide modeling (Leitinger et al., 2008; Maggioni et al., 2016).

Bartelt et al. (2012) developed a two dimensional visco-elastic continuum model to specify the start of gliding snow avalanches. Ancey and Bain (2015) summarized the knowledge concerning the formation of snow glide avalanches and its impact on obstacles in the path. They concluded that meteorological conditions and topographic features causing snow gliding are well known, but the mechanisms are poorly understood.

Feistl et al. (2014) documented the vegetation cover, vegetation height and the terrain properties of 101 glide-snow avalanche release areas on the Dorfberg and indicated four characteristic types of vegetation. Leitinger et al. (2008) established a measure for vegetation roughness (i.e. surface roughness) and showed that this factor has a significant influence on snow-glide distances. However, detailed consideration of the soil-vegetation system in the snow-glide process is missing. This was also noted by Höller (2014) who stated that the conditions at the snow-soil interface have to be investigated most notably.

Although the impact of global change on land cover was mainly due to socio-economic drivers (Tasser et al., 2017), future impact of changing climate will accelerate changes in vegetation composition and vegetation roughness. Hence, studies on causal links and quantitative impacts are especially crucial for snow-gliding and related processes. Besides measures to simplify the complex interactions of vegetation roughness at the snow-ground interface (i.e. surface roughness, Leitinger et al., 2008), the influence of vegetation composition and liquid water to the interlocking of the soil-vegetation-snow continuum is widely unknown.

This study specifically addresses the role of the soil-vegetation system on snow gliding, with an elaborate experimental setup. The focus was on the presence of liquid water in the snowpack, on the vegetation, the soil surface, and in the upmost soil layers, as well as vegetation composition and its consequence on snow gliding. Therefore, these key questions are addressed:

- Which variables in the soil-vegetation system, the snowpack, and the lowest atmospheric boundary layer have considerable influence on snow gliding?
- Is it appropriate to distinguish between processes at the beginning of the winter (development of the snowpack) and the late winter (decline of the snowpack)?
- How does vegetation composition influence the snow gliding process?

## 2 Experimental test site and methods

### 2.1 Test site

The study site is located on the orographic left, south-facing slope of the upper Pinzgau Valley. From the geological point of view, it is a very homogenous area made up mainly of paragneiss and mica schist. This siliceous bedrock is responsible for the presence of cambisols on the pastures. The abandoned and unused areas are mostly based on cambic podzols. The climate at the Wildkogel can be characterized as a subalpine European climate. Long-term average annual rainfall (at 1973 m a.s.l., Schmittenhöhe) amounts to 1501 mm, with the highest monthly precipitations falling in June and August (175–200 mm per month). Long-term average annual temperature is 1.9°C, with the highest monthly average in August at around 10°C. These low temperatures, high precipitation, and the long period of snow cover impose limits on the vegetation period. The investigated slope faces SSE, with slope angles from 20° to 37°.

The area is characterized by pastures and abandoned areas in the immediate vicinity (Baumgärtner, 2016). This situation allowed a comparative approach to be used (Fig. 1).

The pasture is stocked with cattle between the end of June and the beginning of September. This area is dominated by grasses and has been classified as Sieversio montanae–Nardetum strictae subassociation typicum (Lüth et al., 2011) with the matgrass (*Nardus stricta*) as dominant species. Management of the abandoned area ceased about 10 years ago. The predominant species

of the area are dwarf shrubs (e.g. *Vaccinium myrtillus, V. vitis–idaea, Calluna vulgaris*) and the evergreen sedge (*Carex sempervirens*).

## 2.2 Measurements and methods

### 2.2.1 Snow gliding

Snow gliding was measured with glide shoes (In der Gand and Zupancic, 1966). The glide shoes were connected to a drum with a wire. Its displacements generated rotations. A rotary switch generated pulses which were counted by HOBO H6 logger units. The date and time of each pulse was stored. One pulse represents a glide distance of 2.6 mm. A detailed description is given by Leitinger et al. (2008). Forty devices (Fig. 1) were installed at randomly selected places with different land use, topographic conditions, and vegetation characteristics in October 2014 (Baumgärtner, 2016).

The initial force required to displace each shoe was measured with a tension spring balance (Pesola Medio 1000 g). The static friction coefficients for all glide shoes were calculated as the ratio of the initial forces and the normal forces. They represent the influence of different vegetation types and different land uses on snow gliding (Leitinger et al., 2008).

### 2.2.2 Meteorology and related snow and soil properties

An automatic weather station recorded air temperature, air humidity (Rotronic MP103), snow depth (Sommer UHZ8), snow temperatures (Sommer AD592c; 0, 5, 50, 100 cm), and global radiation (Schenk 8101). It was located at the test site. The data were stored at intervals of 10 minutes by a data logger.

At the meteorological station a snow melt analyzer (SMA, Sommer) was available. It measures the dielectric coefficients with a time-domain reflectometer, using two frequencies along a flat band cable. The different dielectric properties of water and ice are used to determine the volume fractions of the LWC and the ice content (Stähli et al., 2004). The flat band cable was mounted 5 cm above the soil surface. It was aligned parallel to the surface and orientated along the fall line. The acquisitions were recorded by a data logger in 10 minute intervals. Data entries were removed in case that the snow depth was less than 5 cm.

Soil temperatures (Pt-100) and soil moistures (Decagon, ECHO®) were measured at four levels (0, 1.5, 5, 10 cm) in the pastures and the abandoned area. The data were stored at intervals of 5 minutes by a data logger (HOBO® Microstation).

### 2.2.3 Topographic features and vegetation characteristics

In order to consider the micro-relief close to the snow glide shoes, topographic features were noted at each glide shoe. The slope angle was measured directly by each glide shoe, as well as one meter uphill and one meter downhill. Along the fall line the distances where the micro relief changed were measured (uphill and downhill). The amplitudes (A) and the wavelength (Λ) of the micro relief were determined. For that purpose, an elastic aluminum pole (length 2 m) was used, which was matched to the ground surface and resulted in a deformation of the slope. With these data, the stagnation depths were calculated according to Salm (1977) for each glide shoe position.

The parameter *static friction coefficient* was determined to estimate the roughness of the vegetation. For calculation, the weight of the glide shoe and the force needed to move the glide shoe on the vegetation surface was measured (Leitinger et al., 2008). A vegetation inventory of each snow gliding measurement plot was made by a simplified phytosociological survey, according to Braun-Blanquet (1964). This involves analyzing the degree to which the important plant species are present at the position of the snow-glide shoes.

To determine phytomass pools at the sites, production analyses were carried out at the beginning of the vegetation period (end of May). Within a harvest frame (size 900 cm²), all above-ground stands were harvested destructively. The experiment consisted of 18 and 22 replicate plots for the pasture and the abandoned/agricultural unused area, respectively.

Knowledge of the absolute amounts of the different functional groups are important in order to assess qualitative vegetation composition and the resulting effects on snow gliding (Newesely et al., 2000; Leitinger et al., 2008). Therefore, the harvested phytomass was divided into the following plant functional groups: grass, herbs, dwarf shrubs, lichens, and mosses. The phytomasses were then oven-dried at 80°C until they reached a constant weight, determined as the dry weight.

## 2.3 Data interpretation and statistical methods

In order to identify the magnitude of the influence of the variables, the snow glide rate is defined as the dependent variable. All other variables are interpreted as independent variables. Since snow gliding in the data set is a binary piece of information for each time step, multiple logistic regression was used to determine the relevant variables (Wilks, 1995). The magnitude of the regression parameters can be used to describe their influence on the dependent variable.

The number of independent variables should be reduced to avoid overfitting. This procedure is often called screening
regression and was established by backward elimination (Wilks, 1995). The procedure starts with all potential predictors. At each step the least important predictor is removed until the termination criteria are reached (tolerance of the predictor >0.2 and variance inflation factor <10).

In about 0.5 % of the data entries snow gliding was recorded. The samples with snow gliding were subsequently weighted. This satisfies that equal amount of 0 and 1 for snow gliding which are used for the multiple logistic regressions (period I: n =
1164096; period II: n = 1340425). A bootstrap is performed by randomly selecting a value, with replacement (i.e. a given value can be represented more than once in the sample). Each sample selected in this manner is used to calculate the regression coefficient B value. This is repeated 100 times, and the generated sample of $B$ values is then used to estimate the standard error and the lower and upper 95% confidence interval. The bootstrapping approach is preferable to that presented by Gude et al. (2009).

The logistic regressions fit the parameters B for all variables. The magnitude of exp(B) is used to describe the intensity of its influence on snow gliding. If exp(B)>1 the effect is positive, which means that the probability of snow gliding rises with increasing values for the variable. Values below 1 have a negative effect, and the probability of snow gliding decreases if the values for the variable rises. exp(B)=1 indicates that the corresponding variable has no influence on snow gliding.

Due to the fact that liquid water at the snow-soil interface is a requirement for intense snow gliding (In der Gand and Zupancic,
1966) the measured soil moisture at 0 cm (soil surface) is analyzed in more detail. By using a multiple linear regression model, the regression coefficient was determined to identify the sign and the magnitude of the independent variables. To avoid overfitting, variables which correlate among themselves were excluded.

In order to consider the differences between the properties of a rising and a degrading snowpack, the data set was divided into two sub-periods: period I from October to January, and period II from February to May. For both periods accuracy tables are
35 used to demonstrate how well the applied method is able to distinguishes between the two classes (gliding, no gliding). As score index the hit rate was used which is the fraction of correctly calculated data records and the sum of all data entries (Wilks, 1995).

The Whitney–Mann U-test is a nonparametric rank test (Schönwiese, 2000). It was used to determine the significance levels (p values) for selected variables.

The statistical analyses were accomplished with the software IBM SPSS Statistics (Version 21, IBM SPSS Statistics Software).

# 3 Results

The time series in Fig. 2 give an overview of the investigated period. The snow cover season started in October 2014 and ended in late May 2015. It was interrupted twice: in November and in May. In period I the soil temperatures decreased until they reached values between 0°C and 1°C. During period II the soil temperatures were nearly constant until the snow melted.

5 At the beginning of the winter (period I) snow gliding was recorded by all glide shoes. The LWC reached more than 4 % (volumetric percent). The soil moisture characteristics were different for pastures and abandoned areas. Though, at the surface, the soil moisture was close to zero until March in both sites (Fig. 2).

At the beginning of period II, the measured LWC values were about 2.5 %. It raised during snow melting, indicated by a rapid decrease in snow height.

## 3.1 Topography and vegetation

An overview of the observations and measurements at the pastures and abandoned areas is given in Tab. 1.

The frequency distributions of vegetation characteristics are L-shaped for all vegetation types (Fig. 3). This indicates that no vegetation type is dominant at the test site. The prevailing slope angle ranges from 25° to 35°. The stagnation depth was below 15 0.5 m, except in one case, indicating a smooth location of that glide shoe. The friction force was low, and in the majority of the cases very low. The frequency distribution of the canopy heights was between 0.01 m and 0.08 m – higher values were less frequent. The distribution of the slope lengths above and below the glide shoes were equally shaped. The distribution of the slope angles below the glide shoes had a maximum at 30°.

## 3.2 Snow gliding

For period I the soil temperature at 10 cm was determined as the variable with the most influence on snow gliding, followed by the LWC (Tab. 2). Moderate influence was detected for soil moisture at 0 cm and soil moisture at 1.5 cm. The soil temperature at 10 cm was the most important variable for period II. A strong negative influence is indicated for the phytomass of mosses and the static friction coefficient in both periods.

The boxplots for the complete data set distinguish between snow gliding and no snow gliding for period I for soil moisture at 0 cm and 1.5 cm (Fig. 4a), soil temperature at 10 cm (Fig. 4c), the LWC (Fig. 4e), the static friction coefficient (Fig. 4g), the phytomass of mosses (Fig. 4i), and the slope length uphill (Fig. 4k). The positive influence of the soil moisture and the soil temperature is obvious, as well as the negative effect of the phytomass of mosses and the static friction coefficient. The influence of LWC on snow gliding exists, but it is low.

In period II the soil moisture at 0 cm and 10 cm (Fig. 4b), the soil temperature at 0 cm and 10 cm (Fig. 4d), the LWC (Fig.4f), the static friction coefficient (Fig. 4h), and the phytomass of mosses (Fig. 4j) affect the snow gliding. The Whitney-Mann U-test shows for all selected variables high significance levels ($p<0.001$).

For period I the hit rate is 85.4 %, and for period II it is 66.0 % (Tab. 3).

## 3.3 Soil water content at 0 cm

In order to determine the relevant variables and quantify their influence on snow gliding, a multiple linear regression was calculated for both the pastures and the abandoned area. The soil moisture at 0 cm was used as the dependent variable. The signs of the regression coefficients indicate a positive or a negative relationship (Tab. 4). The magnitude represents the intensity of its influence on the soil moisture at 0 cm. For both areas, the soil moisture at 10 cm is identified as the most important

variable. Negative correlations were found for soil temperature at 10 cm and snow temperature at 5 cm. Atmospheric variables had a very low influence on the soil moisture at 0 cm.

**4 Discussion and conclusions**

Ceaglio et al. (2017) investigated the role of the soil in the context of snow gliding and the formation of glide cracks and
avalanches. They concluded that the thermal and hydraulic processes in the soil have to be considered. Our study confirms that the soil moisture at the soil surface, and a few centimeters below the surface, are variables which influence the snow glide rates. Additionally, we found that temperatures in the soil have a significant influence on snow gliding. For example, in the second half of December 2014 a freezing process was observed where the soil moisture abruptly decreased after a cold period with no snow on the ground. Furthermore, the phytomass of mosses affects the snow glide rates at the test site.

Clarke and McClung (1999) introduced the terms cold-temperature events and warm-temperature events, which indicate a correlation of glide snow avalanches with air temperatures. Since glide snow avalanches did not occur at the study site, such classification is not useful here. However, to consider different processes during the development of the snowpack and the decline of the snowpack, two sub-periods were defined (period I: October–January; period II: February–May). The soil moisture and the soil temperature had a significant influence on snow gliding in both periods. This indicates a lower viscosity
of the moist snowpack and a water transport from the snowpack towards the soil surface. However, the LWC is not the predominant variable that explains the soil moisture at 0 cm (Tab. 4). Dreier et al. (2016) investigated the influence of meteorological parameters on snow glide avalanches and divided the winter season into two periods. They found that warm temperature events were mostly associated with a melting snow surface, and cold temperature events are linked with hydraulic process in the basal snow layers and the uppermost soil layers. It confirms the conclusions regarding glide distances presented
here.

Some topographical factors also affect snow gliding. In particular, the static friction coefficient has a negative effect on snow gliding. It seems that the friction is reduced by the vegetation, which was depressed by the weight of the snowpack. This depends on the composition and the characteristics of the vegetation (Leitinger et al., 2008). At the test site it can be concluded that dwarf shrubs are more resistant against depression than pastures.

The results also show that the vegetation has a significant effect on snow gliding. Just the phytomass of mosses had a negative influence on snow gliding in both periods. The analyses of the vegetation composition have shown that a higher percentage of mosses exists at low canopy heights (p=-0.52**). Moss-rich and short-stemmed canopies seem to be more interconnected with the snowpack, and thus contribute to a reduction in snow gliding. On the other hand, long-stemmed, grass-rich canopies can be easily felled, and they form an ideal gliding horizon. These findings are in accordance with those of Newesely et al. (2000)
showing that the gliding distances are increasing from cut meadows to pastures to uncut or abandoned grasslands. Furthermore, a canopy height is positively correlated with the proportion of dwarf shrub phytomass (p = -0.73***). The predominant dwarf shrub species in the study area are *Vaccinium sp.* and *Rhododendron ferrugineum*, and are highly lignified and rigid dwarf shrubs. Such dwarf shrubs, as well as small trees, keep the snowpack back and thus reduce snow gliding (see also Newesely et al., 2000; Leitinger et al., 2008). On the other side, the canopy height is negatively correlated with the phytomass of grasses
(p = -0.61***) which promotes snow-gliding (Newesely et al., 2000).

Implications for agricultural land management (Tasser and Tappeiner, 2002) are given as the type of land-use (mowing, grazing), as well as the intensity of land-use (frequency of annual mowing, fertilisation, irrigation, number of grazing animals), lead to characteristic vegetation communities. Mowing and a low level of fertilisation greatly favour the growth of herbs and high growing grasses, while *Nardus stricta* spreads rapidly on meadows with low land-use intensities (usually mown once a
40   year, not fertilised). After land abandonment, *Carex sp.* immediately spreads, forming the climax vegetation at the higher altitudes. Below the natural timberline, however, the proliferation of dwarf shrubs and subsequent a natural reforestation are

taking place. Land-cover changes, especially the transitional forms between meadows of high land-use intensity and young forests may have crucial impact for the snow-gliding process (Newesely et al., 2000; Leitinger et al., 2008). If an adequate land-use intensity cannot be maintained, steep areas have to be reforested to shorten a critical time period of high snow-gliding activity.

5 These investigations on snow gliding confirmed findings from previous studies, and extended them by considering variables describing the vegetation. It seems that the use of soil moisture sensors makes sense for further investigation, which may be focused on the hydraulic processes close to the soil surface. However, upcoming measurement problems of the uppermost partially frozen soil layers must be considered.

## 5 Acknowledgements

10 Snow and weather data including the data of the SMA are part of the operational avalanche warning service of Salzburg (Austria). The authors thank Norbert Altenhofer for providing these data sets, and Michael Butschek for the technical support. We also thank Janette Walde for assistance concerning the statistical work, and Matthias Kammerlander for the support in the field. The helpful comments of two anonymous reviewers improved the article considerably; thanks for their serious contributions.

15 Georg Leitinger and Erich Tasser are part of the Interdisciplinary Research Center 'Ecology of the Alpine Region' within the major research focus 'Alpine Space – Man and Environment' at the University of Innsbruck.

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

**Table 1: Key characteristics of pastures and abandoned/agricultural unused areas. For each land-use type the glide distance and all topographic and vegetation factors are given (mean ± s.e.).**

| Land use | Pasture | Abandoned area |
|---|---|---|
| N | 18 | 22 |
| static friction coefficient ( )* | 0.0 ± 0.0 | 0.1 ± 0.0 |
| stagnation depth (cm) | 16.0 ± 3.6 | 10.2 ± 4.9 |
| slope inclination (°) | 25.0 ± 1.2 | 31.7 ± 1.1 |
| slope inclination uphill (°) | 25.0 ± 1.1 | 29.7 ± 1.6 |
| slope inclination downhill (°) | 31.8 ± 2.5 | 30.2 ± 1.7 |
| slope length uphill (m) | 2.7 ± 0.6 | 1.7 ± 0.4 |
| slope length downhill (m) | 4.6 ± 0.4 | 4.2 ± 0.4 |
| slope orientation (°) | 190.0 ± 0.0 | 186.6 ± 1.9 |
| canopy height (cm) | 2.8 ± 0.3 | 3.7 ± 0.3 |
| cover of dwarf shrubs (%) | 7.6 ± 1.8 | 43.0 ± 6.3 |
| cover of grasses (%) | 28.9 ± 3.5 | 4.7 ± 0.9 |
| cover of herbs (%) | 0.6 ± 0.1 | 0.4 ± 0.1 |
| cover of lichens (%) | 0.4 ± 0.1 | 9.4 ± 1.6 |
| cover of mosses (%) | 0.3 ± 0.1 | 0.3 ± 0.1 |
| phytomass of dwarf shrubs (g m$^{-2}$) | 179.7 ± 28.7 | 739.0 ± 49.7 |
| phytomass of grasses (g m$^{-2}$) | 870.9 ± 24.1 | 156.6 ± 43.8 |
| phytomass of herbs (g m$^{-2}$) | 27.7 ± 6.1 | 26.5 ± 13.9 |
| phytomass of lichens (g m$^{-2}$) | 18.0 ± 5.8 | 178.0 ± 33.9 |
| phytomass of mosses (g m$^{-2}$) | 14.8 ± 4.1 | 6.0 ± 2.0 |
| glide distance (mm) | 144.8 ± 67.1 | 161.1 ± 89.9 |

Table 2. Significant parameters without multi-collinearity and exp(B) of two logistic linear regressions for both periods with snow gliding as dependent variable. If exp(B)<1 then the correlation is negative, if exp(B)>1 then it is positive (bold = most relevant variables, indicated by a difference >0.06 from 1). Bootstrap results are based on 100 bootstrap samples.

5  B = regression coefficient B, s.e. = standard error, exp(B) = odds ratio

| | Without multi-collinearity | | Period I | | | | | | Period II | | | | | |
|---|---|---|---|---|---|---|---|---|---|---|---|---|---|---|
| | Tolerance of the predictor | Variance inflation | exp(B) | sig. | B | s.e. | upper | lower | exp(B) | sig. | B | s.e. | upper | lower |
| soil temperature 0 cm | .731 | 1.368 | 1.015 | 0.000 | .015 | .001 | .013 | .017 | **.809** | **0.000** | **-.212** | **.003** | **-.218** | **-.207** |
| soil temperature 10 cm | .492 | 2.031 | **1.788** | **0.000** | **.581** | **.002** | **.577** | **.586** | **1.352** | **0.000** | **.301** | **.004** | **.294** | **.309** |
| soil moisture 0 cm | .355 | 2.819 | **1.242** | **0.000** | **.216** | **.002** | **.212** | **.220** | **.907** | **0.000** | **-.097** | **.001** | **-.098** | **-.096** |
| soil moisture 1.5 cm | .196 | 5.102 | **1.061** | **0.000** | **.059** | **.001** | **.057** | **.061** | 1.044 | 0.000 | .043 | .001 | .041 | .044 |
| soil moisture 10 cm | .267 | 3.749 | .991 | 0.000 | -.009 | .001 | -.011 | -.007 | **1.110** | **0.000** | **.104** | **.001** | **.103** | **.105** |
| snow height | .495 | 2.021 | 1.013 | 0.000 | .013 | .000 | .013 | .013 | 1.002 | 0.000 | .002 | .000 | .001 | .002 |
| LWC | .421 | 2.376 | **1.390** | **0.000** | **.329** | **.003** | **.323** | **.335** | **1.078** | **0.000** | **.075** | **.002** | **.070** | **.078** |
| air temperature | .353 | 2.836 | 1.035 | 0.000 | .034 | .001 | .032 | .037 | .981 | 0.000 | -.019 | .001 | -.020 | -.018 |
| relative humidity | .554 | 1.804 | 1.009 | 0.000 | .009 | .000 | .008 | .009 | 1.002 | 0.000 | .002 | .000 | .002 | .002 |
| global radiation | .867 | 1.153 | 1.001 | 0.000 | .001 | .000 | .001 | .001 | 1.001 | 0.000 | .001 | .000 | .001 | .001 |
| static friction coefficient | .807 | 1.239 | **.448** | **0.000** | **-.802** | **.033** | **-.873** | **-.731** | **.321** | **0.000** | **-1.137** | **.023** | **-1.177** | **-1.078** |
| stagnation depth | .395 | 2.529 | .998 | 0.000 | -.002 | .000 | -.002 | -.001 | .995 | 0.000 | -.005 | .000 | -.006 | -.005 |
| slope angle | .630 | 1.587 | 1.016 | 0.000 | .016 | .001 | .015 | .018 | 1.060 | 0.000 | .058 | .000 | .058 | .059 |
| slope angle 1 m uphill | .721 | 1.387 | .998 | 0.000 | -.002 | .001 | -.003 | -.001 | .983 | 0.000 | -.017 | .000 | -.018 | -.017 |
| slope angle 1 m downhill | .809 | 1.236 | 1.000 | 0.000 | .000 | .000 | .000 | .001 | .994 | 0.000 | -.006 | .000 | -.007 | -.006 |
| slope length uphill | .631 | 1.584 | **.827** | **0.000** | **-.190** | **.002** | **-.194** | **-.186** | 1.035 | 0.000 | .035 | .001 | .032 | .037 |
| slope length downhill | .790 | 1.266 | 1.008 | 0.000 | .008 | .002 | .005 | .012 | .955 | 0.000 | -.046 | .001 | -.048 | -.044 |
| friction force drum | .392 | 2.553 | 1.009 | 0.000 | .009 | .000 | .009 | .009 | .999 | 0.000 | -.001 | .000 | -.001 | -.001 |
| phytomass of dwarf shrubs | .547 | 1.828 | .988 | 0.000 | -.012 | .000 | -.013 | -.012 | .995 | 0.000 | -.005 | .000 | -.005 | -.005 |
| phytomass of mosses | .752 | 1.330 | **.618** | **0.000** | **-.482** | **.008** | **-.497** | **-.464** | **.355** | **0.000** | **-1.037** | **.006** | **-1.049** | **-1.024** |
| cover of lichen | .583 | 1.715 | .985 | 0.000 | -.016 | .000 | -.016 | -.015 | .980 | 0.000 | -.020 | .000 | -.021 | -.020 |

**Table 3. Contingency table for both periods as a result of the logistic regression.**

|  |  | Snow gliding observed | | | | | |
|---|---|---|---|---|---|---|---|
|  |  | Period I | | | Period II | | |
|  |  | yes | no | Percentage Correct | yes | no | Percentage Correct |
| Snow gliding calculated | yes | 483565 | 99587 | 82.9 | 449104 | 221001 | 67.0 |
|  | no | 70490 | 510454 | 87.9 | 235200 | 435120 | 64.9 |
| Overall Percentage | | | | 85.4 | | | 66.0 |

**Table 4. Regression coefficients of the multiple linear regression, with soil moisture 0 cm as dependent variable.**

|  | Regression coefficients | |
|---|---|---|
|  | Abandoned area | Pastures |
| soil temperature 0 cm | -0.048 | - |
| soil temperature 10 cm | -0.276 | -0.230 |
| soil moisture 5 cm | - | 0.342 |
| soil moisture 10 cm | **0.770** | **0.431** |
| snow temperature 0 cm | 0.189 | 0.234 |
| snow temperature 5 cm | -0.044 | -0.129 |
| snow height | 0.186 | -0.010 |
| LWC | 0.124 | 0.117 |
| air temperature | 0.095 | 0.097 |
| relative humidity | 0.103 | 0.027 |
| global radiation | -0.012 | -0.033 |
| $R^2$ | 0.878 | 0.712 |

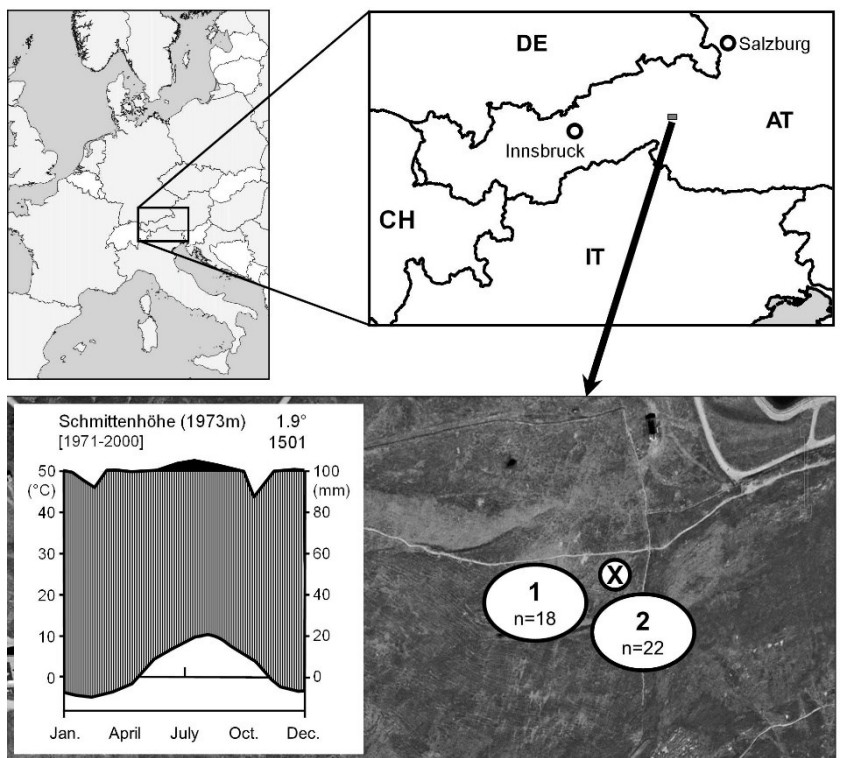

Figure 1. The study area, Wildkogel (Upper Pinzgau, Austria), is characterized by pastures (1) and abandoned areas (2). X = automatic weather station. Original data for the climate diagram: www.zamg.ac.at.

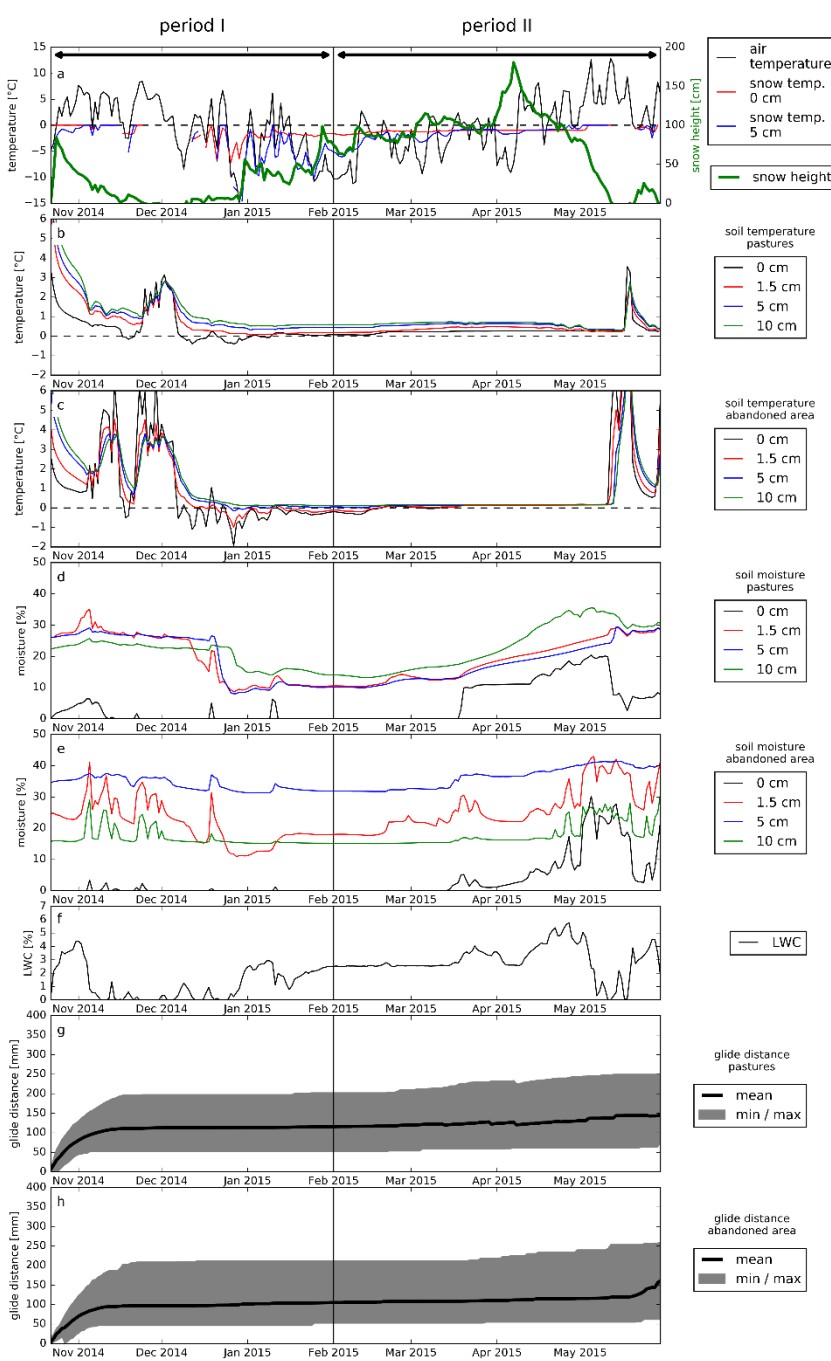

**Figure 2. Time series of meteorological data, soil climate data, and snow properties.**

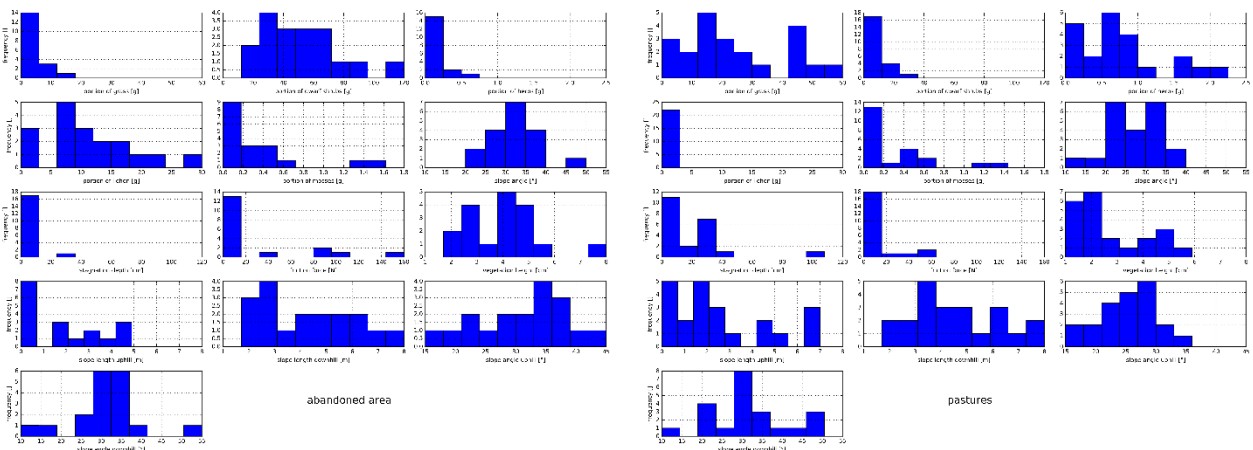

abandoned area

pastures

**Figure 3. Histograms of topographic properties and vegetation characteristics at the glide shoes.**

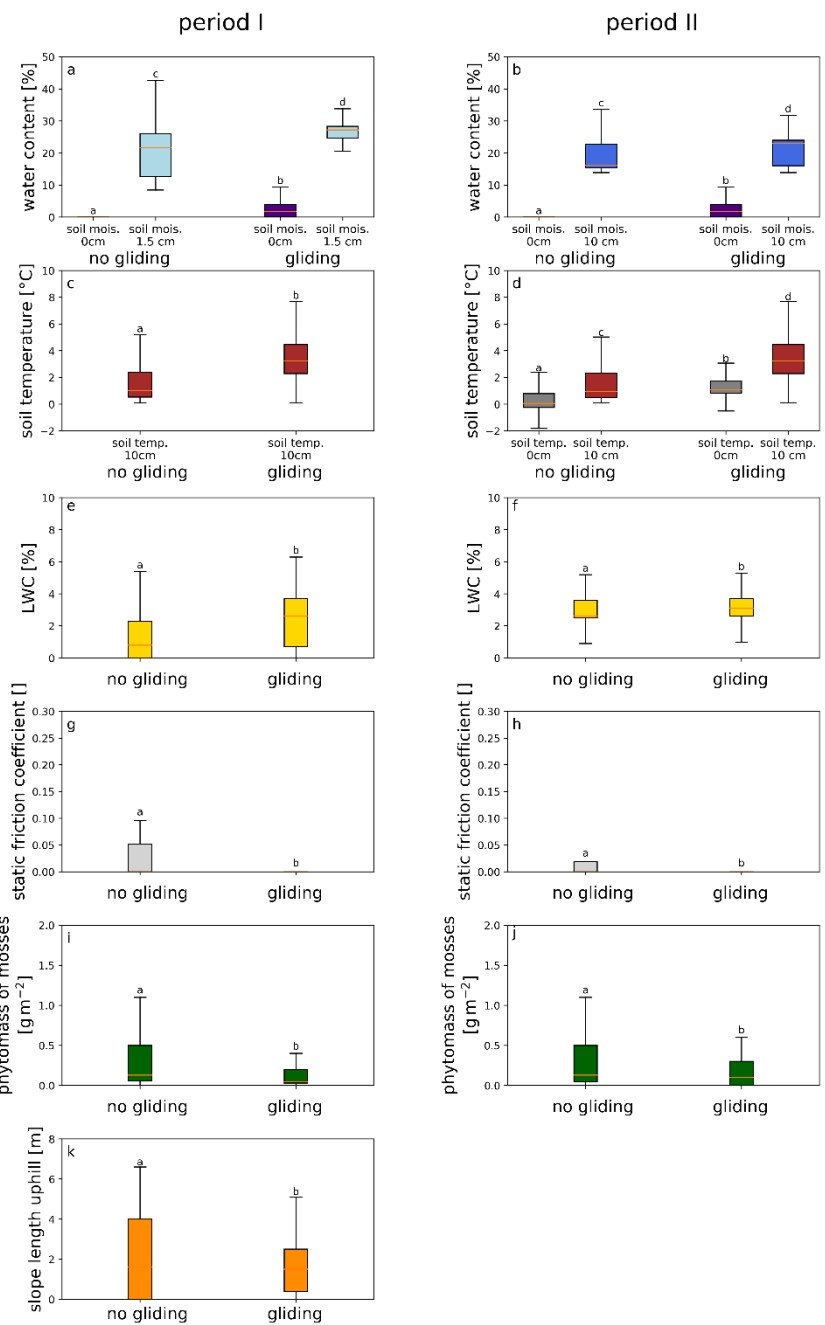

**Figure 4. Boxplots of the most relevant variables in period I and period II (selected according to Tab. 2, indicated by a difference >0.06 of exp(B) from 1). Significant differences between the groups are given by different letters and were determined by Whitney–Mann U-test.**

# 6 Appendix

Correlation matrix between all independent variables

| | 1 | 2 | 3 | 4 | 5 | 6 | 7 | 8 | 9 | 10 | 11 | 12 | 13 | 14 | 15 | 16 | 17 | 18 | 19 | 20 | 21 | 22 | 23 | 24 | 25 | 26 |
|---|---|---|---|---|---|---|---|---|---|---|---|---|---|---|---|---|---|---|---|---|---|---|---|---|---|---|
| 1 soil temperature 0 cm | | 0.90 | 0.69 | 0.59 | 0.04 | 0.25 | 0.10 | 0.31 | 0.54 | -0.39 | -0.39 | -0.26 | -0.38 | 0.27 | 0.09 | 0.05 | 0.00 | 0.00 | 0.09 | 0.05 | 0.00 | -0.01 | -.002* | -0.07 | 0.07 | 0.00 |
| 2 soil temperature 1.5 cm | 0.90 | | 0.90 | 0.83 | -0.01 | 0.32 | 0.16 | 0.35 | 0.39 | -0.48 | -0.40 | -0.20 | -0.39 | 0.23 | 0.20 | 0.02 | 0.01 | -0.04 | 0.07 | 0.08 | 0.01 | 0.01 | -0.01 | -0.02 | 0.06 | 0.02 |
| 3 soil temperature 5 cm | 0.69 | 0.90 | | 0.98 | -0.04 | 0.36 | 0.22 | 0.36 | 0.22 | -0.55 | -0.39 | -0.11 | -0.36 | 0.15 | 0.28 | 0.00 | 0.01 | -0.08 | 0.04 | 0.10 | 0.01 | 0.04 | -0.01 | 0.04 | 0.04 | 0.03 |
| 4 soil temperature 10 cm | 0.59 | 0.83 | 0.98 | | -0.07 | 0.36 | 0.23 | 0.35 | 0.15 | -0.55 | -0.37 | -0.08 | -0.34 | 0.12 | 0.30 | -0.01 | 0.01 | -0.09 | 0.03 | 0.11 | 0.02 | 0.04 | -0.01 | 0.06 | 0.03 | 0.03 |
| 5 soil moisture 0 cm | 0.04 | -0.01 | -0.04 | -0.07 | | 0.60 | 0.09 | 0.73 | 0.43 | -0.25 | -0.22 | 0.04 | -0.19 | 0.56 | 0.02 | 0.03 | 0.04 | -0.01 | 0.14 | 0.12 | -0.03 | 0.02 | -0.06 | -0.11 | 0.11 | 0.03 |
| 6 soil moisture 1.5 cm | 0.25 | 0.32 | 0.36 | 0.36 | 0.60 | | 0.75 | 0.52 | 0.40 | -0.46 | -0.29 | 0.04 | -0.25 | 0.53 | 0.19 | 0.02 | -0.03 | -0.03 | -0.21 | -0.08 | -0.01 | 0.12 | 0.00 | 0.20 | -0.22 | -0.03 |
| 7 soil moisture 5 cm | 0.10 | 0.16 | 0.22 | 0.23 | 0.09 | 0.75 | | -0.03 | 0.15 | -0.10 | .002* | 0.10 | 0.02 | 0.23 | 0.10 | 0.05 | -0.09 | 0.04 | -0.48 | -0.32 | 0.02 | 0.16 | 0.05 | 0.36 | -0.47 | -0.11 |
| 8 soil moisture 10 cm | 0.31 | 0.35 | 0.36 | 0.35 | 0.73 | 0.52 | -0.03 | | 0.36 | -0.53 | -0.36 | 0.03 | -0.31 | 0.52 | 0.16 | 0.03 | 0.08 | -0.09 | 0.29 | 0.27 | -0.05 | -0.01 | -0.07 | -0.16 | 0.26 | 0.08 |
| 9 snow temperature 0 cm | 0.54 | 0.39 | 0.22 | 0.15 | 0.43 | 0.40 | 0.15 | 0.36 | | -0.42 | -0.57 | -0.50 | -0.58 | 0.51 | -0.11 | 0.12 | 0.02 | 0.02 | 0.04 | 0.02 | 0.01 | 0.04 | -0.04 | -0.04 | 0.04 | 0.00 |
| 10 snow height | -0.39 | -0.48 | -0.55 | -0.55 | -0.25 | -0.46 | -0.10 | -0.53 | -0.42 | | 0.79 | 0.27 | 0.74 | -0.33 | -0.24 | 0.05 | -0.07 | 0.12 | -0.12 | -0.15 | 0.01 | -0.02 | 0.04 | -0.01 | -0.13 | -0.05 |
| 11 ice content | -0.39 | -0.40 | -0.39 | -0.37 | -0.22 | -0.29 | .002* | -0.36 | -0.57 | 0.79 | | 0.71 | 0.99 | -0.23 | -0.11 | 0.09 | -0.05 | 0.06 | -0.10 | -0.08 | 0.01 | -0.01 | 0.06 | 0.03 | -0.12 | -0.02 |
| 12 LWC | -0.26 | -0.20 | -0.11 | -0.08 | 0.04 | 0.04 | 0.10 | 0.03 | -0.50 | 0.27 | 0.71 | | 0.78 | 0.00 | 0.02 | 0.08 | -0.03 | -.002* | -0.04 | 0.00 | 0.03 | 0.00 | 0.07 | 0.02 | -0.04 | 0.00 |
| 13 snow density | -0.38 | -0.39 | -0.36 | -0.34 | -0.19 | -0.25 | 0.02 | -0.31 | -0.58 | 0.74 | 0.99 | 0.78 | | -0.21 | -0.09 | 0.09 | -0.05 | 0.06 | -0.10 | -0.07 | 0.01 | -0.01 | 0.07 | 0.03 | -0.12 | -0.02 |
| 14 air temperature | 0.27 | 0.23 | 0.15 | 0.12 | 0.56 | 0.53 | 0.23 | 0.52 | 0.51 | -0.33 | -0.23 | 0.00 | -0.21 | | -0.41 | 0.23 | 0.02 | 0.03 | 0.04 | 0.06 | -0.06 | 0.05 | 0.01 | -0.03 | 0.01 | -0.01 |
| 15 relative humidity | 0.09 | 0.20 | 0.28 | 0.30 | 0.02 | 0.19 | 0.10 | 0.16 | -0.11 | -0.24 | -0.11 | 0.02 | -0.09 | -0.41 | | -0.18 | 0.03 | -0.06 | 0.02 | 0.04 | 0.02 | -0.01 | -0.04 | 0.03 | 0.03 | 0.07 |
| 16 global radiation | 0.05 | 0.02 | 0.00 | -0.01 | 0.03 | 0.02 | 0.05 | 0.03 | 0.12 | 0.05 | 0.09 | 0.08 | 0.09 | 0.23 | -0.18 | | -0.01 | 0.01 | 0.01 | -0.01 | 0.01 | 0.01 | 0.02 | 0.00 | 0.02 | 0.01 |
| 17 friction coefficient | 0.00 | 0.01 | 0.01 | 0.01 | 0.04 | -0.03 | -0.09 | 0.08 | 0.02 | -0.07 | -0.05 | -0.03 | -0.05 | 0.02 | 0.03 | -0.01 | | -0.21 | -0.11 | 0.20 | -0.03 | 0.21 | -0.09 | 0.15 | 0.10 | 0.81 |
| 18 stagnation depth | 0.00 | -0.04 | -0.08 | -0.09 | -0.01 | -0.03 | 0.04 | -0.09 | 0.02 | 0.12 | 0.06 | -.002* | 0.06 | 0.03 | -0.06 | 0.01 | -0.21 | | 0.00 | -0.30 | -0.21 | -0.06 | 0.06 | -0.61 | 0.28 | -0.24 |
| 19 slope angle | 0.09 | 0.07 | 0.04 | 0.03 | 0.14 | -0.21 | -0.48 | 0.29 | 0.04 | -0.12 | -0.10 | -0.04 | -0.10 | 0.04 | 0.02 | -0.01 | -0.11 | 0.00 | | 0.20 | 0.05 | -0.38 | 0.06 | -0.37 | 0.31 | 0.02 |
| 20 slope angle 1 m uphill | 0.05 | 0.08 | 0.10 | 0.11 | 0.12 | -0.08 | -0.32 | 0.27 | 0.02 | -0.15 | -0.08 | 0.00 | -0.07 | 0.06 | 0.04 | -0.01 | 0.20 | -0.30 | 0.20 | | -0.02 | 0.11 | -0.11 | 0.08 | 0.18 | 0.22 |
| 21 slope angle 1 m downhill | 0.00 | 0.01 | 0.01 | 0.02 | -0.03 | -0.01 | 0.02 | -0.05 | 0.01 | 0.01 | 0.01 | 0.03 | 0.01 | -0.06 | 0.02 | 0.01 | -0.03 | -0.21 | 0.05 | -0.02 | | 0.15 | 0.01 | -0.03 | 0.04 | 0.03 |
| 22 slope length uphill | -0.01 | 0.01 | 0.04 | 0.04 | 0.02 | 0.12 | 0.16 | -0.01 | 0.04 | -0.02 | -0.01 | 0.00 | -0.01 | 0.05 | -0.01 | 0.01 | 0.21 | -0.06 | -0.38 | 0.11 | 0.15 | | -0.37 | 0.02 | 0.13 | 0.15 |
| 23 slope length downhill | -.002* | -0.01 | -0.01 | -0.01 | -0.06 | 0.00 | 0.05 | -0.07 | -0.04 | 0.04 | 0.06 | 0.07 | 0.07 | 0.01 | -0.04 | 0.02 | -0.09 | 0.06 | 0.06 | -0.11 | 0.01 | -0.37 | | 0.01 | -0.08 | -0.13 |
| 24 exposition | -0.07 | -0.02 | 0.04 | 0.06 | -0.11 | 0.20 | 0.36 | -0.16 | -0.04 | -0.01 | 0.03 | 0.02 | 0.03 | -0.03 | 0.03 | 0.00 | 0.15 | -0.61 | -0.37 | 0.08 | -0.03 | 0.02 | 0.01 | | -0.68 | 0.18 |
| 25 friction force drum | 0.07 | 0.06 | 0.04 | 0.03 | 0.11 | -0.22 | -0.47 | 0.26 | 0.04 | -0.13 | -0.12 | -0.04 | -0.12 | 0.01 | 0.03 | 0.02 | 0.10 | 0.28 | 0.31 | 0.18 | 0.04 | 0.13 | -0.08 | -0.68 | | 0.11 |
| 26 friction force field | 0.00 | 0.02 | 0.03 | 0.03 | 0.03 | -0.03 | -0.11 | 0.08 | 0.00 | -0.05 | -0.02 | 0.00 | -0.02 | -0.01 | 0.07 | 0.01 | 0.81 | -0.24 | 0.02 | 0.22 | 0.03 | 0.15 | -0.13 | 0.18 | 0.11 | |
| 27 canopy high | 0.05 | 0.10 | 0.16 | 0.18 | 0.05 | -0.05 | -0.23 | 0.21 | -0.01 | -0.18 | -0.08 | 0.04 | -0.07 | 0.00 | 0.08 | 0.01 | 0.07 | -0.45 | -0.10 | 0.45 | 0.38 | 0.24 | -0.09 | 0.28 | 0.13 | 0.17 |
| 28 phytomass of dwarf shrubs | 0.09 | 0.14 | 0.19 | 0.20 | 0.11 | -0.12 | -0.44 | 0.37 | 0.01 | -0.28 | -0.17 | -0.02 | -0.15 | 0.04 | 0.10 | -0.01 | 0.23 | -0.44 | 0.18 | 0.40 | -0.05 | 0.06 | -0.12 | 0.26 | 0.21 | 0.22 |
| 29 phytomass of grasses | -0.11 | -0.12 | -0.12 | -0.12 | -0.13 | 0.27 | 0.64 | -0.40 | 0.01 | 0.23 | 0.17 | 0.07 | 0.16 | -0.02 | -0.09 | 0.03 | -0.06 | -0.02 | -0.56 | -0.31 | 0.27 | 0.41 | 0.09 | 0.23 | -0.29 | -0.04 |
| 30 phytomass of herbs | -0.05 | -0.09 | -0.13 | -0.14 | -0.08 | 0.00 | 0.17 | -0.22 | -0.03 | 0.16 | 0.07 | -0.02 | 0.06 | -0.08 | -0.04 | 0.01 | -0.25 | 0.51 | 0.05 | -0.33 | 0.01 | -0.14 | -0.01 | -0.50 | 0.24 | -0.27 |
| 31 phytomass of lichens | 0.06 | 0.08 | 0.09 | 0.09 | 0.08 | -0.19 | -0.46 | 0.27 | 0.01 | -0.18 | -0.12 | -0.02 | -0.11 | 0.01 | 0.07 | -0.03 | 0.47 | -0.43 | 0.13 | 0.31 | 0.15 | -0.15 | 0.02 | 0.28 | -0.03 | 0.43 |
| 32 phytomass of mosses | 0.00 | -.002* | 0.00 | 0.00 | -0.02 | 0.04 | 0.08 | -0.05 | 0.00 | 0.00 | -0.01 | 0.00 | -0.01 | 0.00 | -0.01 | 0.01 | 0.03 | -0.27 | 0.05 | 0.00 | -0.09 | -0.13 | 0.15 | 0.28 | -0.20 | 0.05 |
| 33 cover of dwarf shrubs | 0.13 | 0.16 | 0.18 | 0.18 | 0.20 | -0.24 | -0.69 | 0.50 | 0.05 | -0.30 | -0.21 | -0.05 | -0.19 | 0.08 | 0.08 | -0.02 | 0.18 | -0.31 | 0.43 | 0.51 | -0.03 | 0.03 | -0.18 | -0.08 | 0.39 | 0.18 |
| 34 cover of grasses | -0.13 | -0.15 | -0.17 | -0.17 | -0.19 | 0.28 | 0.74 | -0.51 | -0.04 | 0.30 | 0.21 | 0.06 | 0.20 | -0.06 | -0.09 | 0.02 | -0.22 | 0.33 | -0.45 | -0.52 | -0.01 | 0.07 | 0.11 | 0.04 | -0.35 | -0.22 |
| 35 cover of herbs | -0.02 | -0.06 | -0.10 | -0.11 | -0.04 | -0.08 | -0.02 | -0.11 | -0.02 | 0.10 | 0.03 | -0.02 | 0.03 | -0.06 | -0.03 | -.002* | -0.20 | 0.70 | 0.13 | -0.18 | -0.04 | -0.18 | 0.00 | -0.64 | 0.47 | -0.23 |
| 36 cover of lichens | 0.05 | 0.07 | 0.08 | 0.08 | 0.05 | -0.17 | -0.39 | 0.21 | -0.01 | -0.14 | -0.08 | -0.01 | -0.08 | -0.01 | 0.06 | 0.00 | 0.23 | -0.38 | 0.18 | 0.25 | 0.17 | -0.24 | 0.15 | 0.28 | -0.13 | 0.25 |
| 37 cover of mosses | -0.04 | -0.05 | -0.05 | -0.05 | -0.08 | 0.11 | 0.26 | -0.18 | -0.02 | 0.07 | 0.03 | -0.02 | 0.02 | -0.04 | -0.02 | -0.01 | -0.11 | -0.05 | 0.00 | -0.10 | -0.11 | -0.26 | 0.14 | 0.22 | -0.35 | -0.11 |