# Peer review of "Determining the drivers for snow gliding"

_Natural Hazards and Earth System Sciences, 2018_

## Referee Comment (RC1) · Anonymous Referee #1 · 7 Feb 2018

**Determining the drivers for snow gliding by Fromm et al. - REVIEW**

The authors present the analyses made on data gathered in an experimental test site for snow gliding, in order to find the most significant drivers for such phenomenon. What is interesting is that, beside the more classical snow and weather variables, they consider as drivers also soil and vegetation.

The topic can be of interest for the readers of NHESS. The paper is worth to be published but only after, I think, major revisions.

In particular, my main concern is about the chosen method for selecting the data to be used for the statistical analyses. The choice might imply some uncertainty in the results which is not discussed in the manuscript. Due to this, at the moment I would like to highlight this fact, without entering too much into other details. Therefore, in the following I report my general comments to the authors and not an exhaustive list of specific comments. I am willing to hear the response of the authors, in order to make a fruitful discussion about this interesting topic and hopefully be helpful in publishing their manuscript.

General comments to the authors:

The manuscript is well written and introduces clearly the argument. The Introduction is very well written.

My main concern is related to the choice of the data used for the analyses and the possible consequences of this choice on the results. You state that "*In about 0.5 % of the data entries snow gliding was recorded. The data set was reduced by randomly selecting data entries without displacements. This satisfies that equal amount of 0 and 1 for snow gliding which are used for the multiple logistic regression.*" (pag. 5, lines 12-14). As in the period of "no gliding" the other parameters (used as independent variables) were very variable (Fig. 2), I think that the results of your analyses might be very different if another random subsample of "no gliding" data was chosen. I think you should try to address this fact, discussing the uncertainty related to the results. Did you try with different subsets?

You should also indicate the number of data in your dataset: 0.5% corresponds to N = ?

Something unclear is also what is the "snow glide rate" that you used as dependent variable? It seems that it takes the value 1 or 0 if there was or not displacement. If this is the case, I would not call it glide rate which includes something related to time (30 min, hourly, daily ?).

Specific comments:

In the Introduction, I think that lines 14-16 (pag. 2) are not needed. Without these lines the section naturally flows to the goal of the manuscript (pag. 2, lines 17-25), where the importance of vegetation appears and is introduced just before (lines 11-13). Lines 14-16 could be little modified and moved to section 2.2.2 (pag. 4).

Table 1 already presents some results. I would move it in section 3. Moreover, in the caption of Table 1 you write "*… For each land-use type the glide distance and all…*", but no glide distance is given.

At the end of section 2.2.3 (pag. 4, line 30) there is a part that should belong to the result section. I would move this part in a new subsection of section 3 related to topography and vegetation. Also Table 1 and the figure of the Appendix should be moved in this new section. I would also make this figure distinguishing between abandoned and pasture areas. Though it is not the main goal of the manuscript, showing the

difference of the vegetation types in the two different plots would anyhow provide useful information for discussion.

Still in section 2.2.3 you write "*The stagnation depth was below 0.5 m, except in one case, indicating a smooth location of that glide shoe.*" (pag. 4, line 32). Apart from this statement, concerning the roughness of the site, you show in Table 1 values for "vegetation roughness" in the pasture and abandoned areas… how did you determine these values? Is this parameter related to stagnation depth? Please describe this or refer to literature.

Pag. 5, lines 23-24. Was the division in period I and II done according to a general rule or to the registered data in your study site? Dreier et al. (2016) and Ceaglio et al. (2017) explicitly write that their choices were based on the specific snow and weather conditions of their study sites. Please, give a reason for your choice, even if kind of expert-based.

In section 3. Results, I would eliminate the first subsection "Time series" and just begin the section with "*The time series…*", then make the more specific subsections 3.2 and 3.3 after the new subsection on topography and vegetation.

In section 3.2 (pag. 6, line 9) you give values for the overall mean glide distance which I cannot find in Fig. 2. What are the values 185.9 and 361.8 mm? In Fig. 2 the black lines should represent the same values at the end of the period, right? Do I miss something? Please, check and explain well this… I would also write somewhere what a "click" in the measuring device for glide distance corresponds to. In Leitinger et al. (2008), which you refer to in section 2.2.1, it seems that it corresponds to 2.6 mm. Is this right?

At pag. 6, lines 1-3 you write something that is not represented in Fig. 2. The soil moisture at 10 cm (green line) in the abandoned area is around 15 %, not zero as you write here. Please check this.

Caption of Fig. 2 is incomplete. You show also snow glide distance… for which it is needed to write what it is the black line and the grey area around that line.

In the boxplot of Fig. 3, did you use the whole dataset or again only the subset which were used in the logistic regression? This is not clear.

And again it is not clear to me how you chose the parameters for the box plots and the Whitney–Mann U-test. In the caption of Table 2 your write "… (*bold = most relevant variables, indicated by a large difference from 1).*", but then some of the values are not much far from 1 (for ex. soil moisture at 1.5 cm) !?!?

At pag. 8 the discussion on snow gliding and vegetation properties is very interesting, but it is strange that some *p* values appear here for the first time without being presented before… did you do some correlation analyses? Why don't you present all the results of the correlation analyses in the results section and then discuss them here?

---

## Referee Comment (RC2) · Anonymous Referee #2 · 19 Feb 2018

Review of manuscript «Determining the drivers for snow gliding» (Fromm et al. )

General comments:

The manuscript aims at determining the drivers for snow gliding under the effect of changing soil moisture conditions (also in relation development vs. decline of snowpack) and vegetation characteristics. The authors found that soil moisture at the soil surface (1$^{st}$ Part of winter) and soil moisture 1.5 cm below the soil (2$^{nd}$ part of winter) were the most important variables. They found also important vegetation effects. The presented work fills thus important research gaps and has the potential to be a valuable contribution to the state of research on snow gliding processes. I see however several points which should be improved before publishing in NHSS, most importantly:

1. The story of the manuscript should be focused more towards answering the three research questions and towards the main conclusions (which are not yet so clear for me). Two of the three research questions are dealing with vegetation effects on snow gliding. So, this topic should be introduced and discussed better in the light of previous work and implications for land-use management.
2. Some methodological aspects should be clarified (see also specific comments). Generally , the methods used in this work have been conducted carefully, but they partly fail at disentangling potentially confounding variables. Surprisingly significant results (e.g. effects of lichens and mosses on snow gliding) should thus be better checked for interactions with other variables or at least carefully discussed before publishing.
3. The form and presentation of the manuscript could be improved in different ways (see also specific comments). Some parts of the text is not yet nicely structured in topical paragraphs. Some sections could be shortened without a loss of relevant information towards the main conclusions. Some captions to figures and tables are not 100% clear. The English language would deserve an additional check.

Specific comments:

For a reviewer it would be helpful to have continuous line numbers in order to refer in the review to a specific text.

Page 1, l. 16. Abstract: was it really the lower phytomass of mosses that had a negative influence on snow gliding or was it not just the lower canopy height of these sites, which was related to phytomass of mosses?

P1, l. 17-18. Did a higher phytomass of dwarf shrubs really reduce snow gliding? According to table 2 I see that exp (B) for this variable is very close to 1 for the 1$^{st}$ period and not given for the 2$^{nd}$ period.

P1, l. 24. The 3$^{rd}$ sentence « Höller summarized the findings.. » is in this from not necessary for the introduction of the research questions.  Please just add the reference where it fits and contributes to the state of research.

P2, l. 11-16 – The paragraph on the role of vegetation is important for the understanding of the manuscript (2 of 3 research questions are dealing with vegetation effects). The paragraph would deserve thus some more attention in the introduction. In the current form the topic is just introduced by the statement that not much is known about vegetation effects (ignoring thus various publications on snow-glide vegetation effects) before the topic is again abruptly changed to LWC in the same paragraph.

P2, l 20-25, research questions: the two first research questions make sense, but the 2$^{nd}$ research question is not really introduced in the preceding introduction. The 3$^{rd}$ question is also relevant, but is in my eyes not really answered here. The manuscript provides some information on the association between snow gliding with different plant types (eg. mosses or lichens), but I can't find information about the effect of different land-use types (e.g. pasture, abandoned land).

Section 2.1: the test-site section is quite long and partly redundant with Fig. 1. Please avoid where possible paragraphs with only 1 sentence (in the whole manuscript). I would also reduce the number of listed plant species (because most readers of NHSS are probably not be familiar with them) and focus on the most characteristic and for snow gliding most relevant dwarf shrub and grass species (or vegetation types). It is not clear from the description of the study area if we have 2 or 3 treatments (is abandoned and unusable the same treatment or not). And are slope angles and other topographical variables the same for the different categories?

Section 2.2.1 : The description of the design of the distribution of the glide shoes is rather vague. How many glide shoes were distributed in pastures vs abandoned land and which other criteria were used to distribute them?

Section 2.2.2 : Some of the very technical information in this section could potentially be shortened without substantial loss of information.

p.4. line 21. Please replace « after Braun-Blanquet » with « according to Braun-Blanquet »

p5, line 12-14. I'm a bit confused by the statement that about 0.5% of the data entries contain snow-gliding and the data set was reduced to have an equal number of snow gliding vs. no snow gliding. I agree that the numbers of 0 and 1 in a logistic model should be similar or at least in the same range, so the approach seems ok for me. But this would means that c. 90% of the data entries without gliding have been thrown away. Could you provide here numbers of data entries with and without snow gliding and the criteria used for this categorization.

p. 6, line 8ff. Was slope angle not a relevant variable or was the variation in slope angle so small? I would have expected also a boxplot with snow-gliding vs. slope angle.

p. 6, line 26, replace "very significant" by "highly significant"

p. 6, line 29-30 (and elsewhere): please avoid where possible method description in the result section

p. 7, line. 13-15. It is not necessary to repeat the objective of the study here. The objective should be clear from the introduction.

p. 7, line 18-19. It is for me a bit surprising that the phytomass of mosses has an influence on snow gliding. While I'm not surprised that you received a significant relationship, I expect mainly a confounding effect between phytomass of mosses and other variables which may have a more direct effect on snow gliding (also indicated on p. 8, line 10, relationship with canopy height). Such potentially confounding relationships are not easy to disentangle with multivariate logistic models alone. I would suggest to check additionally for such relationships or at least to discuss such a result (which is also repeated in the abstract) and potential confounding effects with other variables

p8, l8: snow gliding or snow sliding?

P8, l. 9 ff. similar case like

Table 1: do the abandoned areas include "unusuable land"? And was there actually a difference in snow gliding for the different land-use types? Do the results of this study confirm earlier studies (e.g. by Leitinger, Tasser et al. ?)

Table 3: the content of the contingency table is interesting but should be better explained in the table caption. The model for period 1 was obviously better than for period 2, which can be interpreted quite well with differences in relevant variables for both periods

Fig. 3: The description of the figure could be clearer.  In the first graph on the left, the y-axis is water content, but there is also a boxplot on soil moisture in the same graph. And what do the A, B, AA, BB mean?

---

## Author Comment (AC1) · 20 Mar 2018

We thank the reviewer for the comments and we address the various concerns below. In this answer we mainly consider questions and remarks from the general comments and major specific comments. Full answers to minor specific comments (mainly wording and rephrasing as well as improving figures and tables) will be included in the revised version of the manuscript. Reviewer comments are highlighted (R), with our response below (A) in each case.

R: [General comments to the authors:]

R:[My main concern is related to the choice of the data used for the analyses and the possible consequences of this choice on the results. You state that "In about 0.5 % of the data entries snow gliding was recorded. The data set was reduced by randomly selecting data entries without displacements. This satisfies that equal amount of 0 and

1 for snow gliding which are used for the multiple logistic regression." (pag. 5, lines 12-14). As in the period of "no gliding" the other parameters (used as independent variables) were very variable (Fig. 2), I think that the results of your analyses might be very different if another random subsample of "no gliding" data was chosen. I think you should try to address this fact, discussing the uncertainty related to the results. Did you try with different subsets? ]

A: We agree that randomly selecting sub-samples may cause variations of the coefficients exp(ß). The magnitude of these variations will be determined by choosing several sets of randomly selected data records for the analysis (i.e. statistical bootstrapping). This will demonstrate the quality of the fit. Table 2 will be extended with the range of the values of exp(ß). We did some tests with several random samples in advance and based on these results we expect small variations with the main correlations/results staying the same. However, the interpretation of variables with exp(ß) close to 1 will be revised based on the new results.

R: [You should also indicate the number of data in your dataset: 0.5% corresponds to N = ?]

A: N = 5259. We will add this information in the revised version of the manuscript.

R: [Something unclear is also what is the "snow glide rate" that you used as dependent variable? It seems that it takes the value 1 or 0 if there was or not displacement. If this is the case, I would not call it glide rate which includes something related to time (30 min, hourly, daily ?). ]

A: Displacements of glide shoes originate electrical pulses which are recorded. A pulse is produced by a rotary switch when the glide shoe moves 2.6 mm. All remaining data (temperature, moisture etc.) are registered in intervals of 10 minutes. Therefore, the snow displacement is calculated for these 10 minute intervals (in millimeter per 10 minutes) for each glide shoe. We will improve the wording in the revised version of the manuscript to avoid confusion.

R: [Specific comments:]

R suggests restructuring some parts in the sections 'methods' and 'results' as well as minor changes of the wording in the 'introduction' section.

A: We will follow these suggestions in order to improve the manuscript. We will clearly distinguish what belongs to which section (methods, results and discussion). A new subsection will collect all information concerning topography and vegetation. The subsection 'time series' will be removed. We will take care to ensure that there are no new terms in the discussion section.

R: [Still in section 2.2.3 you write "The stagnation depth was below 0.5 m, except in one case, indicating a smooth location of that glide shoe." (pag. 4, line 32). Apart from this statement, concerning the roughness of the site, you show in Table 1 values for "vegetation roughness" in the pasture and abandoned areas... how did you determine these values? Is this parameter related to stagnation depth? Please describe this or refer to literature.]

A: We have realized that terminology is not distinct regarding the different measures of roughness (vegetation versus ground). We will clarify this issue and improve wording to avoid confusion.

R: [In section 3.2 (pag. 6, line 9) you give values for the overall mean glide distance which I cannot find in Fig. 2. What are the values 185.9 and 361.8 mm? In Fig. 2 the black lines should represent the same values at the end of the period, right? Do I miss something? Please, check and explain well this... I would also write somewhere what a "click" in the measuring device for glide distance corresponds to. In Leitinger et al. (2008), which you refer to in section 2.2.1, it seems that it corresponds to 2.6 mm. Is this right?]

A: Due to graphical representation, Figure 2 is currently not showing all the data used in the statistics as some data loggers have stopped logging due to a full memory. We

will clarify this issue and update Fig. 2 (mean, max, min snow gliding distance) to be in accordance with the data used in statistics. You are right about the distance for one "click" in the measuring device, it corresponds to 2.6 mm. We will add this information as already indicated in the general comments.

R: [At pag. 8 the discussion on snow gliding and vegetation properties is very interesting, but it is strange that some p values appear here for the first time without being presented before... did you do some correlation analyses? Why don't you present all the results of the correlation analyses in the results section and then discuss them here?]

A: Thank you for this comment and the positive evaluation of our discussion on snow gliding and vegetation. We apologize for the confusion. In the revised version of the manuscript we will ensure that all results are presented in the appropriate section and nothing new will be presented in the discussion section.

A: Thank you for your valuable comments, which will significantly improve our manuscript. We are looking forward to present you a revised version of our manuscript.

Please also note the supplement to this comment:
https://www.nat-hazards-earth-syst-sci-discuss.net/nhess-2018-3/nhess-2018-3-AC1-supplement.pdf

---

## Author Comment (AC2) · 20 Mar 2018

We thank the reviewer for the comments and we address the various concerns below. In this answer we mainly consider questions and remarks from the general comments as well as major specific comments. However, we would include answers to all specific comments in a revised version of the manuscript. Reviewer comments are highlighted (R), with our response below (A) in each case.

R: [The story of the manuscript should be focused more towards answering the three research questions and towards the main conclusions (which are not yet so clear for me). Two of the three research questions are dealing with vegetation effects on snow gliding. So, this topic should be introduced and discussed better in the light of previous work and implications for land-use management.]

A: We agree that the research questions raised at the end of the introduction are not

adequately answered in the conclusion section. We will integrate more aspects of previous studies concerning vegetation and snow gliding. Especially, we will look more closely at the differences in vegetation (reported among others by Newesely et al. (2000) and Meusburger et al. (2014). Former investigations have shown that the vulnerability of alpine ecosystems to snow gliding increases with the reduction of agricultural use. High snow gliding rates were observed on low but soft dwarf shrub canopies which is an early stage in the secondary succession after abandonement. Referring to these findings, we will improve our 'introduction' as well as 'discussion' section.

R: [Some methodological aspects should be clarified (see also specific comments). Generally, the methods used in this work have been conducted carefully, but they partly fail at disentangling potentially confounding variables. Surprisingly significant results (e.g. effects of lichens and mosses on snow gliding) should thus be better checked for interactions with other variables or at least carefully discussed before publishing.]

A: Some variable names are not used consistently throughout. This leads to confusion in some parts of the manuscript and makes the manuscript difficult to comprehend. We will eliminate these flaws in a revised version of the manuscript. There are collinearities between some variables. This results from statistical analyses which are described and applied. To allow better interpretation of confounding effects as well as correlations, a correlation matrix for all used independent variables will be provided as supplemental material. Moreover, we will improve discussion on most valuable confounding effects in the revised version of the manuscript.

R: [The form and presentation of the manuscript could be improved in different ways (see also specific comments). Some parts of the text is not yet nicely structured in topical paragraphs. Some sections could be shortened without a loss of relevant information towards the main conclusions. Some captions to figures and tables are not 100% clear. The English language would deserve an additional check.]

A: The sections 'methods', 'results' and 'discussion' will be reorganized. The paragraphs describing the test site and the snow gliding conditions will be assigned to the methods section. The section 'results' will then only contain results and we will take care to ensure that there are no new terms/results are shown in the discussion section. We will revise figure and table captions following your specific comments.

R: [Specific comments]

R: [P2, l. 11-16 – The paragraph on the role of vegetation is important for the understanding of the manuscript (2 of 3 research questions are dealing with vegetation effects). The paragraph would deserve thus some more attention in the introduction. In the current form the topic is just introduced by the statement that not much is known about vegetation effects (ignoring thus various publications on snow-glide vegetation effects) before the topic is again abruptly changed to LWC in the same paragraph.]

A: We will integrate more aspects of previous studies concerning vegetation and snow gliding and better represent this main topic of our study in the 'introduction' section (as already indicated in our answer to your general comments).

R: [P2, l 20-25, research questions: the two first research questions make sense, but the 2nd research question is not really introduced in the preceding introduction. The 3rd question is also relevant, but is in my eyes not really answered here. The manuscript provides some information on the association between snow gliding with different plant types (eg. mosses or lichens), but I can't find information about the effect of different land-use types (e.g. pasture, abandoned land).]

A: Throughout revision we will slightly adapt our research questions to the improved introductory part regarding snow gliding and vegetation impact. We will also add available information about different land-use types associated with variations of plant types.

R: [Section 2.1: the test-site section is quite long and partly redundant with Fig. 1. Please avoid where possible paragraphs with only 1 sentence (in the whole manuscript). I would also reduce the number of listed plant species (because most

readers of NHSS are probably not be familiar with them) and focus on the most characteristic and for snow gliding most relevant dwarf shrub and grass species (or vegetation types). It is not clear from the description of the study area if we have 2 or 3 treatments (is abandoned and unusable the same treatment or not). And are slope angles and other topographical variables the same for the different categories? Section 2.2.1 : The description of the design of the distribution of the glide shoes is rather vague. How many glide shoes were distributed in pastures vs abandoned land and which other criteria were used to distribute them?]

A: We will thoroughly revise figures and tables following your valuable comments. Information on spatial distribution of snow glide shoes as well as numbers will be added to Figure 1.

R: [p5, line 12-14. I'm a bit confused by the statement that about 0.5% of the data entries contain snowgliding and the data set was reduced to have an equal number of snow gliding vs. no snow gliding. I agree that the numbers of 0 and 1 in a logistic model should be similar or at least in the same range, so the approach seems ok for me. But this would means that c. 90% of the data entries without gliding have been thrown away. Could you provide here numbers of data entries with and without snow gliding and the criteria used for this categorization.]

A: We have used a random sample of '0' values to be in accordance with case numbers of '1' values. Also requested from Reviewer #1, we will determine the magnitude of variations by choosing several sets of randomly selected data records for the analysis (i.e. statistical bootstrapping). This will demonstrate the quality of the fit. Table 2 will be extended with the range of the values of exp(ß). We did some tests with several random samples in advance and based on these results we expect small variations with the main correlations/results staying the same. However, the interpretation of variables with exp(ß) close to 1 will be revised based on the new results.

R:[p. 7, line 18-19. It is for me a bit surprising that the phytomass of mosses has an

influence on snow gliding. While I'm not surprised that you received a significant relationship, I expect mainly a confounding effect between phytomass of mosses and other variables which may have a more direct effect on snow gliding (also indicated on p. 8, line 10, relationship with canopy height). Such potentially confounding relationships are not easy to disentangle with multivariate logistic models alone. I would suggest to check additionally for such relationships or at least to discuss such a result (which is also repeated in the abstract) and potential confounding effects with other variables p8, l8: snow gliding or snow sliding?]

A: Thank you for emphasizing the importance of better explanation/interpretation of confounding variables. Although potential confounding relationships are not easily to detect, we will provide a correlation matrix of all involved variables as supplement as well as a more comprehensive discussion of the main confounding effects in the discussion section in the revised version of our manuscript.

Please also note the supplement to this comment:
https://www.nat-hazards-earth-syst-sci-discuss.net/nhess-2018-3/nhess-2018-3-AC2-supplement.pdf

---

## Author Response (AR1)

**ANSWERS to the review #1**

**Determining the drivers for snow gliding by Fromm et al.**

Reviewer comments and questions are black and answers are green.

The authors present the analyses made on data gathered in an experimental test site for snow gliding, in order to find the most significant drivers for such phenomenon. What is interesting is that, beside the more classical snow and weather variables, they consider as drivers also soil and vegetation.

The topic can be of interest for the readers of NHESS. The paper is worth to be published but only after, I think, major revisions.

In particular, my main concern is about the chosen method for selecting the data to be used for the statistical analyses. The choice might imply some uncertainty in the results which is not discussed in the manuscript. Due to this, at the moment I would like to highlight this fact, without entering too much into other details. Therefore, in the following I report my general comments to the authors and not an exhaustive list of specific comments. I am willing to hear the response of the authors, in order to make a fruitful discussion about this interesting topic and hopefully be helpful in publishing their manuscript.

General comments to the authors:

The manuscript is well written and introduces clearly the argument. The Introduction is very well written.

My main concern is related to the choice of the data used for the analyses and the possible consequences of this choice on the results. You state that "*In about 0.5 % of the data entries snow gliding was recorded. The data set was reduced by randomly selecting data entries without displacements. This satisfies that equal amount of 0 and 1 for snow gliding which are used for the multiple logistic regression.*" (pag. 5, lines 12-14). As in the period of "no gliding" the other parameters (used as independent variables) were very variable (Fig. 2), I think that the results of your analyses might be very different if another random subsample of "no gliding" data was chosen. I think you should try to address this fact, discussing the uncertainty related to the results. Did you try with different subsets?

We agree that randomly selecting sub-samples cause variations of the coefficients exp(β). Therefore, we performed bootstrapping to demonstrate the consequences (100 times). An additional paragraph in sub-section 2.3 explains the approach. In table 3 the range of value B is shown by implementing the lower and upper limits of B.

You should also indicate the number of data in your dataset: 0.5% corresponds to N = ?

This part was slightly modified due to the requested bootstrap analysis and the numbers are now provided: "The samples with snow gliding were subsequently weighted. This satisfies that equal amount of 0 and 1 for snow gliding which are used for the multiple logistic regressions (period I: n = 1164096; period II: n = 1340425)."

Something unclear is also what is the "snow glide rate" that you used as dependent variable? It seems that it takes the value 1 or 0 if there was or not displacement. If this is the case, I would not call it glide rate which includes something related to time (30 min, hourly, daily ?).

Displacements of glide shoes originate electrical pulses which are recorded. A pulse is produced by a rotary switch when the glide shoe moves 2.6 mm.
All remaining data (temperature, moisture etc.) are registered in intervals of 10 minutes. Therefore, the snow displacement is calculated for these 10 minute intervals (in millimeter per 10 minutes) for each glide shoe. We will improve the wording in the revised version of the manuscript to avoid confusion.

Specific comments:

In the Introduction, I think that lines 14-16 (pag. 2) are not needed. Without these lines the section naturally flows to the goal of the manuscript (pag. 2, lines 17-25), where the importance of vegetation appears and is introduced just before (lines 11-13). Lines 14-16 could be little modified and moved to section 2.2.2 (pag. 4).

The lines 14-16 (pag. 2) are deleted. A is added sentence in section 2.2.2 which indicates that the SMA sensor was already used in a study concerning the triggering of wet-snow avalanches.

Table 1 already presents some results. I would move it in section 3. Moreover, in the caption of Table 1 you write "*… For each land-use type the glide distance and all…*", but no glide distance is given.

Table 1 is moved to section 3 in a new sub-section 3.1. as proposed.

We added a row in Table 1 named "glide distances".

At the end of section 2.2.3 (pag. 4, line 30) there is a part that should belong to the result section. I would move this part in a new subsection of section 3 related to topography and vegetation. Also Table 1 and the figure of the Appendix should be moved in this new section. I would also make this figure distinguishing between abandoned and pasture areas. Though it is not the main goal of the manuscript, showing the difference of the vegetation types in the two different plots would anyhow provide useful information for discussion.

A new sub-section 3.1. "Topography and vegetation" is added. It contains Table 1 which is moved from section 2.2.3. and the figure from the appendix with the histograms of topographic properties and vegetation characteristics. Now, we distinguish between abandoned and pasture areas. Hence, the figure numbering is adjusted.

Still in section 2.2.3 you write "*The stagnation depth was below 0.5 m, except in one case, indicating a smooth location of that glide shoe.*" (pag. 4, line 32). Apart from this statement, concerning the roughness of the site, you show in Table 1 values for "vegetation roughness" in the pasture and abandoned areas… how did you determine these values? Is this parameter related to stagnation depth? Please describe this or refer to literature.

Sorry for this confusion. The static friction coefficient is the measure for roughness of the vegetation and calculated according to Leitinger et al. (2008). We have deleted "vegetation roughness" from Tab. 1 and added description how we determined the static friction coefficient.

Pag. 5, lines 23-24. Was the division in period I and II done according to a general rule or to the registered data in your study site? Dreier et al. (2016) and Ceaglio et al. (2017) explicitly write that their choices were based on the specific snow and weather conditions of their study sites. Please, give a reason for your choice, even if kind of expert-based.

Major snow gliding was observed in autumn (at the beginning of the winter snowpack) and in spring (during intense melting). We decided to separate the two periods. Therefore, the decision is expert-based.

In section 3. Results, I would eliminate the first subsection "Time series" and just begin the section with "*The time series…*", then make the more specific subsections 3.2 and 3.3 after the new subsection on topography and vegetation.

We followed these suggestions and removed the sub-title "Time series". The new subsection 3.1 contain Tab. 1 which gives an overview of the conditions and characterizes the test site.

In section 3.2 (pag. 6, line 9) you give values for the overall mean glide distance which I cannot find in Fig. 2. What are the values 185.9 and 361.8 mm? In Fig. 2 the black lines should represent the same values at the end of the period, right? Do I miss something? Please, check and explain well this… I would also write somewhere what a "click" in the measuring device for glide distance corresponds to. In Leitinger et al. (2008), which you refer to in section 2.2.1, it seems that it corresponds to 2.6 mm. Is this right?

The glide distances are correct now. The end of the time series used in this study is the end of May (Fig. 2). In the previous manuscript we used the latest data entry in logger from June when we removed the devices from the field.

In order to avoid duplications in the manuscript, the first line in sub-section 3.2 (snow gliding) is removed. The glide distances of pastures and abandoned areas are integrated in Tab. 1.

The information that one pulse represents a glide distance of 2.6 mm is added in sub-section 2.2.1.

At pag. 6, lines 1-3 you write something that is not represented in Fig. 2. The soil moisture at 10 cm (green line) in the abandoned area is around 15 %, not zero as you write here. Please check this.

This was an error. We used the wrong column, first. The text is modified.

Caption of Fig. 2 is incomplete. You show also snow glide distance… for which it is needed to write what it is the black line and the grey area around that line.

The black line represents the mean glide distance. The gray area indicates the range between the minimum and maximum values. Now, this information is added in the legend.

In the boxplot of Fig. 3, did you use the whole dataset or again only the subset which were used in the logistic regression? This is not clear.

The whole data set was used for the box plots. In order to communicate that to the reader we expanded the introducing sentence to the histograms: "The boxplots for the complete data set …"

And again it is not clear to me how you chose the parameters for the box plots and the Whitney–Mann Utest. In the caption of Table 2 your write "… (*bold = most relevant variables, indicated by a large difference from 1).*", but then some of the values are not much far from 1 (for ex. soil moisture at 1.5 cm) !?!?

Table 2 is revised and extended: bold = most relevant variables, indicated by a difference >0.05 from 1. Bootstrapping is applied and the results are based on 100 bootstrap samples.

At pag. 8 the discussion on snow gliding and vegetation properties is very interesting, but it is strange that some *p* values appear here for the first time without being presented before… did you do some correlation analyses? Why don't you present all the results of the correlation analyses in the results section and then discuss them here?

The Whitney-Mann U-test and its corresponding p values are now introduced in the methods section. The correlation matrix is added in the appendix.
* * *
**ANSWERS to the review #2**

**«Determining the drivers for snow gliding» (Fromm et al.)**

Reviewer comments and questions are black and answers are blue.

General comments:

The manuscript aims at determining the drivers for snow gliding under the effect of changing soil moisture conditions (also in relation development vs. decline of snowpack) and vegetation characteristics. The authors found that soil moisture at the soil surface (1st Part of winter) and soil moisture 1.5 cm below the soil (2nd part of winter) were the most important variables. They found also important vegetation effects. The presented work fills thus important research gaps and has the potential to be a valuable contribution to the state of research on snow gliding processes. I see however several points which should be improved before publishing in NHSS, most importantly:

1. The story of the manuscript should be focused more towards answering the three research questions and towards the main conclusions (which are not yet so clear for me). Two of the three research questions are dealing with vegetation effects on snow gliding. So, this topic should be introduced and discussed better in the light of previous work and implications for land-use management.

Thank you for this comment. We now introduced especially the state-of-the-art and gap of knowledge regarding the vegetation effects on snow gliding in the Introduction section. The results are discussed better and implications for land-use management are presented in the discussion section.

2. Some methodological aspects should be clarified (see also specific comments). Generally , the methods used in this work have been conducted carefully, but they partly fail at disentangling potentially confounding variables. Surprisingly significant results (e.g. effects of lichens and mosses on snow gliding) should thus be better checked for interactions with other variables or at least carefully discussed before publishing.

We now address this topic (as indicated in the specific comments) by stating the influence and relevance of confounding variables. By both checking and discussing this topic we clarify the mentioned methodological aspects.

3. The form and presentation of the manuscript could be improved in different ways (see also specific comments). Some parts of the text is not yet nicely structured in topical paragraphs. Some sections could be shortened without a loss of relevant information towards the main conclusions. Some captions to figures and tables are not 100% clear. The English language would deserve an additional check.

We improved the structure of the manuscript. A new sub-section "Topography and vegetation" is created in section 3 (results). It contains Table 1 which is moved from section 2.2.3. and the figure from the appendix with the histograms of topographic properties and vegetation characteristics. Now, we distinguish between abandoned and pasture areas. Hence, the figure numbering is adjusted. Furthermore, the headline of sub-section "Time series" is removed (section 3).

Some captions of figures and tables are written in more detail.

The manuscript has been professionally proofread to ensure correct grammar and spelling.

Specific comments:

For a reviewer it would be helpful to have continuous line numbers in order to refer in the review to a specific text.

The template from NHESS was used for the submission of the manuscript. We kindly ask the editor to forward this suggestion to *Copernicus Publications*.

Page 1, l. 16. Abstract: was it really the lower phytomass of mosses that had a negative influence on snow gliding or was it not just the lower canopy height of these sites, which was related to phytomass of mosses?

Thank you for this comment. You are right! Not the lower phytomass of mosses reduces the snow gliding, but the simultaneously increased canopy height. We introduced this point now in the Abstract.

P1, l. 17-18. Did a higher phytomass of dwarf shrubs really reduce snow gliding? According to table 2 I see that exp (B) for this variable is very close to 1 for the 1$^{st}$ period and not given for the 2$^{nd}$ period.

The results show that dwarf shrub coverage has a significant negative impact in period 2 (see also Table 2, exp (B) = 0.88).

P1, l. 24. The 3rd sentence « Höller summarized the findings.. » is in this from not necessary for the introduction of the research questions. Please just add the reference where it fits and contributes to the state of research.

The sentence was deleted.

P2, l. 11-16 – The paragraph on the role of vegetation is important for the understanding of the manuscript (2 of 3 research questions are dealing with vegetation effects). The paragraph would deserve thus some more attention in the introduction. In the current form the topic is just introduced by the statement that not much is known about vegetation effects (ignoring thus various publications on snow-glide vegetation effects) before the topic is again abruptly changed to LWC in the same paragraph.

We have expanded our reasoning regarding the vegetation effects on snow gliding in the Introduction section. The results are now discussed in this light and implications for land-use management are presented in the discussion section.

P2, l 20-25, research questions: the two first research questions make sense, but the 2nd research question is not really introduced in the preceding introduction. The 3rd question is also relevant, but is in my eyes not really answered here. The manuscript provides some information on the association between snow gliding with different plant types (eg. mosses or lichens), but I can't find information about the effect of different land-use types (e.g. pasture, abandoned land).

We rephrased the 3rd research question to be in line with our main findings and the analyses on different plant types (i.e. plant functional groups) on the snow gliding process. The 2nd research question is now introduced in the Introduction section.

Section 2.1: the test-site section is quite long and partly redundant with Fig. 1. Please avoid where possible paragraphs with only 1 sentence (in the whole manuscript). I would also reduce the number of listed plant species (because most readers of NHSS are probably not be familiar with them) and focus on the most characteristic and for snow gliding most relevant dwarf shrub and grass species (or vegetation types). It is not clear from the description of the study area if we have 2 or 3 treatments
(is abandoned and unusable the same treatment or not). And are slope angles and other topographical variables the same for the different categories?

We improved the description of our test-site and reduced the number of listed plant species to the most abundant ones. In the description of the study area as well as improved Fig. 1 we clarified the experimental setup and number of treatments.

Section 2.2.1: The description of the design of the distribution of the glide shoes is rather vague. How many glide shoes were distributed in pastures vs abandoned land and which other criteria were used to distribute them?

Table 1 as well as Fig. 1 contains the number of glide shoes in pastures (18) and abandoned areas (22).

No additional rules or criteria were applied to choose their locations. We added "… randomly selected places …" in sub-section 2.2.1.

Section 2.2.2: Some of the very technical information in this section could potentially be shortened without substantial loss of information.

The sub-section is shortened by removing sentences containing low information or sentences are reformulated.

p.4. line 21. Please replace « after Braun-Blanquet » with « according to Braun-Blanquet »

The suggestion has been implemented.

p5, line 12-14. I'm a bit confused by the statement that about 0.5% of the data entries contain snowgliding and the data set was reduced to have an equal number of snow gliding vs. no snow gliding. I agree that the numbers of 0 and 1 in a logistic model should be similar or at least in the same range, so the approach seems ok for me. But this would means that c. 90% of the data entries without gliding have been thrown away. Could you provide here numbers of data entries with and without snow gliding and the criteria used for this categorization.

We have now calculated the statistics via a bootstrapping and rephrased the whole section to clarify our methodical approach. 'In about 0.5 % of the data entries snow gliding was recorded. The samples with snow gliding were subsequently weighted. This satisfies that equal amount of 0 and 1 for snow gliding which are used for the multiple logistic regressions (period I: n = 1164096; period II: n = 1340425). A bootstrap is performed by randomly selecting a value, with replacement (i.e. a given value can be represented more than once in the sample). Each sample selected in this manner is used to calculate the regression coefficient B value. This is repeated 100 times, and the generated sample of *B* values is then used to estimate the standard error and the lower and upper 95% confidence interval. The bootstrapping approach is preferable to that presented by Gude et al. (2009).'

p. 6, line 8ff. Was slope angle not a relevant variable or was the variation in slope angle so small? I would have expected also a boxplot with snow-gliding vs. slope angle.

The mean and the standard deviation of the variable "slope angle" is shown in Tab. 1 and its relevance given in Tab. 2. We found that it is a significant variable for snow gliding, but its influence is low. Other studies with higher variations of the slope angels in their test sites investigated its role in more detail.

p. 6, line 26, replace "very significant" by "highly significant"

The suggestion has been implemented.

p. 6, line 29-30 (and elsewhere): please avoid where possible method description in the result section

The accuracy tables and the Whitney-Mann U-test (including p values) are now introduced in the methods section.

p. 7, line. 13-15. It is not necessary to repeat the objective of the study here. The objective should be clear from the introduction.

The sentence is deleted.

p. 7, line 18-19. It is for me a bit surprising that the phytomass of mosses has an influence on snow gliding. While I'm not surprised that you received a significant relationship, I expect mainly a confounding effect between phytomass of mosses and other variables which may have a more direct effect on snow gliding (also indicated on p. 8, line 10, relationship with canopy height). Such potentially confounding relationships are not easy to disentangle with multivariate logistic models alone. I would suggest to check additionally for such relationships or at least to discuss such a result (which is also repeated in the abstract) and potential confounding effects with other variables

Thank you for this comment. We have revised the entire manuscript to clarify such relationships and have inserted a correlation matrix (Appendix) to support our statements.

p8, l8: snow gliding or snow sliding?

This typing error was corrected in the whole manuscript.

P8, l. 9 ff. similar case like

Table 1: do the abandoned areas include "unusuable land"? And was there actually a difference in snow gliding for the different land-use types? Do the results of this study confirm earlier studies (e.g. by Leitinger, Tasser et al. ?)

There is no significant difference between the abandoned areas and "unusable land". Both sites are currently not managed, but we do not know for sure from one site whether it was used in former times. Therefore we have introduced this subdivision; however, we have deleted the term 'unusable land' in order to avoid confusion.
The results confirm earlier studies (see Discussion section).

Table 3: the content of the contingency table is interesting but should be better explained in the table caption. The model for period 1 was obviously better than for period 2, which can be interpreted quite well with differences in relevant variables for both periods

The table caption is extended. And the percentage for each class is given now. This facilitates the interpretation.

Fig. 3: The description of the figure could be clearer.  In the first graph on the left, the y-axis is water content, but there is also a boxplot on soil moisture in the same graph. And what do the A, B, AA, BB mean?

[revised manuscript text omitted]
 | 0.532.495 | 2.021 | 1.878013 | 0.000 | .013 | .000 | .013 | .013 | 1.006002 | 0.000 | -.002 | -.000 | .001 | .002 |
| LWC | 0.522.421 | 2.376 | 1.916390 | 0.000 | .329 | .003 | .323 | .335 | 1.405078 | 0.000 | -.075 | -.002 | .070 | .078 |
| air temperature | .353 | 2.836 | 1.035 | 0.2890 00 | 3.455.0 34 | -.001 | -.032 | -.037 | -.981 | 0.000 | -.019 | .001 | -.020 | -.018 |
| relative humidity | 0.542.554 | 1.845804 | 1.006009 | 0.000 | -.009 | -.000 | .008 | .009 | 1.002 | 0.000 | .002 | .000 | .002 | .002 |
| global radiation | 0.876.867 | 1.141153 | -1.001 | -0.000 | .001 | .000 | .001 | .001 | 1.001 | 0.01200 00 | .001 | .000 | .001 | .001 |
| static friction coefficient | 0.296.807 | 3.3731.239 | .448- | -0.000 | 0.060-.802 | .0330.0 00 | -.873 | -.731 | .321 | 0.000 | -1.137 | .023 | -1.177 | -1.078 |
| stagnation depth | 0.392.395 | 2.553529 | 1.008.9 98 | 0.000 | 1.017-.002 | 0.000 | -.002 | -.001 | .995 | 0.000 | -.005 | .000 | -.006 | -.005 |
| slope angle | 0.609.630 | 1.643587 | 1.026 16 | 0.000 | .0164.0 35 | 0.001 | .015 | .018 | 1.060 | 0.000 | .058 | .000 | .058 | .059 |
| slope angle 1 m uphill | 0.693.721 | 1.442387 | 0.881.9 98 | 0.000 | -.002 | -.001 | -.003 | -.001 | .983 | 0.000 | -.017 | .000 | -.018 | -.017 |
| slope angle 1 m downhill | 0.787.809 | 1.270236 | -1.000 | -0.000 | -.000 | -.000 | .000 | .001 | .994 | 0.000 | -.006 | .000 | -.007 | -.006 |
| slope length uphill | 0.538.631 | 1.860584 | -.827 | -0.000 | -.190 | -.002 | -.194 | -.186 | 1.035 | 0.000 | .035 | .001 | .032 | .037 |
| slope length downhill | 0.784.790 | 1.276266 | -1.008 | -0.000 | -.008 | -.002 | .005 | .012 | .955 | 0.000 | -.046 | .001 | -.048 | -.044 |
| friction force drum | 0.378.392 | 2.644553 | 1.009 | 0.000 | 1.009 | 0.000 | .009 | .009 | .999 | 0.000 | -.001 | .000 | -.001 | -.001 |

| | | | | | | | | | | | | | |
|---|---|---|---|---|---|---|---|---|---|---|---|---|---|
| friction force field | 0.311 | 3.213 | 0.996 | 0.000 | - | - | | | | | | | |
| Phytomassphytomass of dwarf shrubs | 0.527.547 | 1.898828 | .9880.994 0.000 | --.012 | -.000 | -.013 | -.012 | .995 0.000 | -.005 | .000 | -.005 | -.005 | |
| phytomass of mosses | 0.250.752 | 1.330 | .618 0.000 | -.482 | 4.008 | 0.425-.497 | -.4640.000 | 0.462.355 0.000 | -1.037 | .006 | -1.049 | -1.024 | |
| cover of lichen | 0.516.583 | 1.939715 | 0.993.985 0.027000 | 0-.016.988 | 0.039.000 | -.016 | -.015 | .980 0.000 | -.020 | .000 | -.021 | -.020 | |
| cover of moss | 0.229 | 4.367 | 1.209 | 0.000 | - | - | | | | | | | |

5   **Table 3. Contingency table for both periods as a result of the logistic regression.**

[revised manuscript text omitted]

---

## Referee Report (RR1)

**Determining the drivers for snow gliding by Fromm et al. – REVIEW – second turn**

The authors made a great work in accomplishing my comments and in answering properly and clearly to them. The paper is now ready for publications, after only some minor revision.

In particular, they still do not say explicitly how they selected the parameters for the box-plots and the Whitney–Mann U-test (Fig. 4). In Table 2 there are 7 significant parameters for Period I and 8 parameters for Period II, but then the box-plots are made only for some of those parameters. Why? Maybe they did it and the results were not significant… no problem, they should simply state this.

At pag. 5, lines 30-33 should go to the method, where actually lines 24-27 at pag. 4 tell more or less the same.

At page. 5, lines 1, I would change the sentence into: "*The soil moisture characteristics were different for pastures and abandoned areas. Though, at the surface, the soil moisture was close to zero until March in both sites (Fig. 2)."*

And here it would be interesting to comment about freezing and melting processes in the uppermost soil layers. In fact, the authors speak about these processes in the abstract and in the conclusion, but without really using their data. But, actually, this fact appears in their data! At the end of December 2014, the soil moisture abruptly decreased after a cold period (Ta down to - 15 °C) with no snow at ground… I would imagine then that this decrease was related to the freezing of the liquid water in the soil. I imagine that at this point of the manuscript the authors might give this example to support their sentences (in the abstract and at the end of the conclusion) about the importance of freezing and melting process in the uppermost soil layers. It is simply a suggestion…

---

## Author Response (AR2)

[revised manuscript text omitted]
 | -0.04 | -0.05 | -0.05 | -0.05 | -0.08 | 0.11 | 0.26 | -0.18 | -0.02 | 0.07 | 0.03 | -0.02 | 0.02 | -0.04 | -0.02 | -0.01 | -0.11 | -0.05 | 0.00 | -0.10 | -0.11 | -0.26 | 0.14 | 0.22 | -0.35 | -0.11 |

**Answer to the Review #2**
**Determining the drivers for snow gliding by Fromm et al. – REVIEW – second turn**

The authors made a great work in accomplishing my comments and in answering properly and clearly to them. The paper is now ready for publications, after only some minor revision.

In particular, they still do not say explicitly how they selected the parameters for the box-plots and the Whitney–Mann U-test (Fig. 4). In Table 2 there are 7 significant parameters for Period I and 8 parameters for Period II, but then the box-plots are made only for some of those parameters. Why? Maybe they did it and the results were not significant… no problem, they should simply state this.

We modified Figure 4. Now boxplots for all variables with significant influence on snow gliding are shown. The rule (the difference of exp(B) from 1) is the same as for table 2. This information is added in the figure caption.

At pag. 5, lines 30-33 should go to the method, where actually lines 24-27 at pag. 4 tell more or less the same.

The sentence is removed.

At page. 5, lines 1, I would change the sentence into: "*The soil moisture characteristics were different for pastures and abandoned areas. Though, at the surface, the soil moisture was close to zero until March in both sites (Fig. 2)."*

The sentence is modified according to the suggestion of the reviewer.

And here it would be interesting to comment about freezing and melting processes in the uppermost soil layers. In fact, the authors speak about these processes in the abstract and in the conclusion, but without really using their data. But, actually, this fact appears in their data! At the end of December 2014, the soil moisture abruptly decreased after a cold period (Ta down to - 15 °C) with no snow at ground… I would imagine then that this decrease was related to the freezing of the liquid water in the soil. I imagine that at this point of the manuscript the authors might give this example to support their sentences (in the abstract and at the end of the conclusion) about the importance of freezing and melting process in the uppermost soil layers. It is simply a suggestion…

Thanks for this suggestion. We considered it and added a sentence in the discussions and conclusions.

We thank the editor for the helpful comments and considered two scientific articles which are related to the topic snow gliding and vegetation.